# Proteomic Insights into the Interaction of Chitosan Nanoparticles with Canine MDCK Epithelial Cells

**DOI:** 10.3390/molecules30183815

**Published:** 2025-09-19

**Authors:** Lorena E. Galván-Flores, Carlos Osorio-Trujillo, Patricia Talamás-Rohana, Salvador Gallardo-Hernández

**Affiliations:** 1Programa de Doctorado en Nanociencias y Nanotecnología, Centro de Investigación y de Estudios Avanzados del Instituto Politécnico Nacional, Av. IPN 2508, Ciudad de Mexico C. P. 07360, Mexico; 2Departamento de Infectómica y Patogénesis Molecular, Centro de Investigación y de Estudios Avanzados del Instituto Politécnico Nacional, Av. IPN 2508, Ciudad de Mexico C. P. 07360, Mexico; clostrujillo2@yahoo.com.mx (C.O.-T.); ptr@cinvestav.mx (P.T.-R.); 3Departamento de Física, Centro de Investigación y de Estudios Avanzados del Instituto Politécnico Nacional, Av. IPN 2508, Ciudad de Mexico C. P. 07360, Mexico

**Keywords:** biomaterials, CS NPs, cell migration, proteins

## Abstract

Chitosan is considered an excellent biomaterial for epithelial healing treatment. However, information on its molecular interaction with cells at a molecular level is still lacking. Thus, in the present study, homemade synthesized chitosan nanoparticles (CS NPs) and their physicochemical characterization were examined; we found that NPs had average sizes of 15, 30, and 125 nanometers by modifying some variables in the synthesis protocols. It is worth noting that a crystalline structure was found on the smallest NPs, with an average size of 15 nm, as observed in high-resolution transmission electron micrographs. To study the in vitro interaction of CS NPs with Madin–Darby canine kidney (MDCK) cells, co-culturing was performed, and cell viability was assessed. We found that NPs were not toxic at concentrations of up to 400 µg/mL during the first 24 h. Additionally, quantitative mass spectrometry revealed the overexpression of several proteins induced by the co-culture of CS NPs with MDCK cells, and a proteomic analysis suggested two important things: possible clathrin-mediated endocytosis could be the interaction mechanism of CS NPS with MDCK, and proteins related to cytoskeleton formation and organization were overexpressed. Moreover, wound healing assays revealed that <125 nm> CS NPs yielded the best closure rates, where mitomycin was added to make sure that only cell migration occurred.

## 1. Introduction

Chitosan is a well-known biopolymer derived from chitin. Its low cost and abundance have promoted its use in several industrial applications. Moreover, due to its biocompatibility, biodegradability, and antimicrobial activity [1], combined with its flexibility in the production of nanoparticles, it has also been widely employed in biomedical research and healthcare. There are at least two main applications in the biomedical research field. The first is the use of CS nanoparticles as nanocarriers for drug delivery systems, offering as many possibilities as there are research fields within nanomedicine. The second is their use in tissue engineering (from which wound healing could be considered a subset or application area within tissue engineering). Despite the large amount of research carried out in this biomedical area for the application of CS at the laboratory scale, CS still needs to be chemically modified at the hydroxyl and free amino groups to overcome its low water solubility and become suitable for the aforementioned biomedical applications [2]. For more information on the biomedical applications of CS NPs, you can refer to comprehensive reviews already available in the literature [3].

From pioneering work on wound healing in dogs’ urogenital tissue in the 1980s [4] due to the accessibility of the skin as a tissue, several developments on micro- and nanofibers based on mixtures of chitosan with other polymers, obtained by electrospinning, have been reported in the literature. However, some other tissues, such as the epithelial tissue from the ovaries, lungs, and gastrointestinal tract, are not equally accessible as the skin. Although the goal was the same—to heal the tissue—the number of published works and the methods for studying epithelial tissues have not been the same as those used in skin studies. One pioneering work proves the permeability of monolayers of intestinal epithelial cells due to chitosan. The authors studied the mechanism of peptide transport enhancement across epithelia by following a marker using confocal microscopy. The authors were particularly interested in nasal epithelium; however, due to difficulties in obtaining confluent cell cultures, they proposed intestinal cells as an alternative model epithelium [5]. Regarding intestinal epithelium, one of the recent interests of researchers is the development of future treatments for celiac disease, in which chitosan has shown some healing effects [6]. Despite careful reviews that have been published attempting to correlate results across different cell lines and chitosan derivatives [7], to ensure the safety of future chitosan treatments, cytotoxicity experiments are mandatory. Following this, further information on the interaction between chitosan and specific cell lines, such as the regulation of proteins, will be required.

MDCK epithelial cells have proven to be helpful in forming monolayers due to their ability to assemble both tight and adherent junctions, induce cell polarity, and exhibit collective motility. It is well known that chitosan (CS) is primarily obtained from the alkaline deacetylation of chitin (poly *N*-acetyl β-1,4-glucosamine), resulting in a random distribution of acetylated and deacetylated units. This cationic polysaccharide is composed of D-glucosamine (2-amino-2-deoxy-β-d-glucopyranose) and *N*-acetylglucosamine (2-acetamide-2-deoxy-β-d-glucopyranose), linked by β glucosidic bonds (1–4). Chitosan has two functional groups in its chemical structure, which determine the degree of deacetylation and its molecular weight: the amino group (-NH_2_) and the hydroxyl group (OH). Due to the presence of amino and hydroxyl groups in the polymer chain, chitosan’s molecular structure can be modified, allowing for its use in drug delivery [8], regeneration, and wound healing [9]; it has mucoadhesive [10,11], anticancer [12], procoagulant [13], antimicrobial [14], and anti-inflammatory properties. To date, published studies have demonstrated the early epithelialization of wounds resulting from the use of chitosan-based mesoscopic dressings or fibers, suggesting that the dressing provides a scaffolding structure conducive to rapid healing [15]. Although numerous studies in the literature address the material’s biocompatibility and wound-healing capabilities, only a few studies have investigated the interaction at the cellular level; however, they lack information about the molecular level [16]; thus, further research is needed to elucidate the interaction mechanism between CS and cells.

This study aimed to characterize homemade chitosan nanoparticles of defined average sizes and evaluate their interaction with epithelial cells. Thus, the study mainly uses mass spectrometry to demonstrate the impact of co-culturing CS NPs with MDCK cells on cell viability and migration, the latter as a consequence of protein overexpression. Moreover, the proteomics developed in this work could be useful in determining which proteins could serve as a starting point for studying the signaling pathways involved in the capture and internalization of chitosan.

## 2. Materials and Methods

### 2.1. Reagents

The materials employed in this work were bulk chitosan with a low molecular weight (Sigma Aldrich, Saint Louis, MO, USA, Cat. No. 448869), acetic acid (JTBaker, Radnor, PA, USA, Cat. No. 950805), and sodium tripolyphosphate (TPP) (Sigma Aldrich, Cat. No. 238503) used as a crosslinking agent for the synthesis of Cs NPs. For the cell culture, DMEM (Dulbecco’s Modified Eagle Medium GIBCO, Waltham, MA, USA, Cat. No. 11965092), FBS (fetal bovine serum GIBCO, Waltham, MA, USA, Cat. No. 16000044), and a mixture of non-essential amino acids (GIBCO, Cat. No. 1974074) were used. Finally, 3-[4,5-dimethylthiazole-2-yl]-2,5-diphenyltetrazolium bromide (MTT) (Sigma-Aldrich, Cat. No. 2060695) was used for cell viability assays, and Alexa Fluor^TM^ 488 Phalloidin (Thermo Fisher Scientific, Waltham, MA, USA, Cat. No. A12379) and Hoechst (Thermo Fisher Scientific, Waltham, MA, USA, Cat. No. 33258) were used for immunofluorescence assays.

### 2.2. Synthesis of CS NPs

Chitosan’s molecular weight was determined to be 164,518.02 Da through liquid chromatography with a diode array detector (HPLC-DAD) using a Perkin Elmer (SERIES 200, Waltham, MA, USA). CS NPs were prepared using ionotropic gelation [17]. First, 1 mg/mL of CS was dissolved in a 1% (*v*/*v*) acetic acid solution for 4 h at 55 °C. The self-assembly of CS nanoparticles was achieved by stirring a solution of CS and sodium tripolyphosphate (TPP) at a volume ratio of 4:1 (CS:TPP). During chemical synthesis, the stirring time and TPP concentration were varied to determine the average size of the nanoparticles (see Table 1); each average size obtained will be named as “protocol”. CS NPs were centrifuged at 14,000 rpm for 15 min and resuspended in milli-Q water to avoid additional toxicity due to residual acetic acid. Finally, CS NPs were filtered through a 0.22 µm membrane to filter the sample that will be in contact with live cells, to avoid particles, microorganisms, and other contaminants larger than 0.22 μm.

### 2.3. Optical and Morphological Characterization of CS NPs

Optical properties like absorption of NPs were determined using UV-vis spectroscopy to verify the formation of CS NPs, as evidenced by the presence of absorbance peaks, explained as molecular orbital transitions. The particle size and morphology of CS NPs were observed using atomic force microscopy (AFM) (NT-MDT, model Solver Next 1100, Moscow, Russia) in semi-contact top mode, with a gain factor of 0.21 and a cantilever frequency of 8 kHz. For high-resolution transmission electron microscopy (TEM) (JOEL-JEM-2010, Peabody, MA, USA), an aliquot of CS NPs was placed onto a copper grid and dried. An acceleration voltage of 200 kV was applied. Measurements were performed using a Zetasizer nano ZSP (Malvern Panalytical, Malvern, UK) in 173° backscatter mode at 25 °C with a 40 μL cuvette [ZEN0040] (Brand, Aachen, Germany). The data were analyzed using Zetasizer software (v.7.12).

All samples were aliquoted from a colloidal suspension of CS NPs with a nominal concentration of 1 µg/µL. To obtain a homogeneous sample, the colloidal suspension was vortexed. For AFM and Raman spectroscopy, an aliquot of 8 µL was left on a substrate (silicon wafer or glass, respectively) for one hour at room temperature until dry. For determining the optical absorption, a PerkinElmer Lambda 25 UV-Vis system (PerkinElmer Inc., Shelton, CT, USA) was used, which has a spectral range of 200–1100 nm. Due to the size of the quartz cell, a 3 mL volume was required. Thus, variations in the sample concentration due to inhomogeneity at the micrometric level were expected. Regarding vibrational characterization, sample measurements were performed using a Hiroba Jobin (Hiroba Jobin Yvon, Kyoto, Japan) Lab Raman spectrometer equipped with an excitation line operating at 632.8 nm and a nominal power of 20 mW from a He-Ne laser.

### 2.4. Cell Culture

Madin–Darby canine kidney cells (donated by Dr. Abigail Betanzos, CINVESTAV) were cultured in Dulbecco’s Modified Eagle Medium (DMEM), supplemented with 5% fetal bovine serum (FBS) and 1% non-essential amino acids, at 37 °C with 5% CO_2_ and humidity. The cells reached 90% confluence in each experiment.

### 2.5. Cell Viability Assay

Cell viability was determined using an MTT (3-[4,5-dimethylthiazole-2-yl]-2,5-diphenyltetrazolium bromide) assay. Cells (1 × 10^4^) were placed in two different 96-well culture plates (one for 24 and the other for 48 h) and incubated with DMEM containing four different concentrations of CS NPs (100, 200, 300, and 400 µg/mL) for 24 and 48 h, with three average nanoparticle sizes (15, 30, and 125 nm). Then, a 20 µL aliquot of MTT solution (5 mg/mL) was added to each well, and the plates were incubated for an additional 4 h at 37 °C. Subsequently, the medium was aspirated, and 100 µL of acid–isopropanol was added to each well. Optical density was measured at a wavelength of 595 nm in a BIO-RAD 680 ELISA microplate reader to determine absorbance. The experiment was performed in triplicate for statistical accuracy.

### 2.6. Protein Extraction

MDCK cells (1 × 10^6^) were seeded in a T-75 flask and incubated overnight until they reached 90% confluency to obtain protein extracts. Subsequently, the cells were treated with two concentrations of CS NPs, 200 and 300 µg/mL, for 8 h. Cell lysis was obtained with RIPA buffer (500 µL) containing 50 mM Tris-HCl at pH 7.4, 150 mM NaCl, 1% NP-40, 1 mM EDTA, and protease inhibitor cocktail 1×. The total extract was obtained as follows: mechanical detachment of the cells, sonication five times for 20 s, and centrifugation at 14,000 rpm for 15 min to separate the supernatant from the pellet. Subsequently, the Lowry method was used for protein quantification. A sample buffer with β-mercaptoethanol was then added to the proteins, and they were boiled in a water bath for 5 min. The samples were loaded onto a 10% acrylamide gel for electrophoresis analysis. The samples were run at a constant voltage of 120 V. Afterward, the gel was stained with Coomassie blue for 30 min, followed by destaining until the proteins were observed.

### 2.7. Mass Spectrometry

A quantitative proteomic analysis was performed on cell samples incubated with 200 µg/mL of CS NPs <125 nm> for 8 h, and 400 µg/mL of CS NPs (<125 nm>) for 48 h in the Genomic, Proteomic, and Metabolomic Unit (UGPM) at Cinvestav using a mass spectrometer (WATERS-SYNAPT G2-SI, Milford, MA, USA). Subsequently, 50 µg of total protein, suspended in RIPA buffer, was precipitated using methanol/chloroform and enzymatically digested using an iST Sample Preparation Kit (PreOmics, Munich, Germany). Then, 50 µL of Lyse reagent was added to the protein pellets, and they were placed in a heating block for 10 min at 95 °C at 1000 rpm. The samples were sonicated using a BioRuptor Pico (Diagenode, Liège, Belgium) with 20 cycles of 30 s ON and 30 s OFF. The protein samples were digested using 50 µL of a Lys-C/Trypsin mix (Digest reagent) and heated at 37 °C for 2 h. The resulting peptides were cleaned in an iST cartridge using Wash 1 buffer to eliminate hydrophobic contaminants and Wash 2 buffer to eliminate hydrophilic contaminant molecules; afterward, the peptides were eluted using Elute reagent and dried in a SpeedVac (TermoFisher Scientific, Waltham, MA, USA). Then, the peptides were resuspended with LC-Load reagent and normalized at a concentration of 1 µg/µL; an aliquot of alcohol dehydrogenase 1 (ADH1) from *Saccharomyces cerevisiae* (Uniprot accession P00330; 1 pmol/µL) was added as an internal standard to obtain a final concentration of 25 fmol/µL. Bioinformatic analyses of mass spectrometry data, including protein–protein interactions and gene ontology, were performed on the STRING [18] and Gene Ontology [19] websites, respectively.

### 2.8. Scratch Wound Healing

Cells (4 × 10^5^) were placed in 24-well plates and cultured overnight (at 90% confluence). Subsequently, the cells were treated with mitomycin in the culture medium (10 µg/mL) for 2 h to inhibit cell proliferation. After this, the mitomycin-containing medium was discarded, and each well was washed three times with 1× PBS buffer prior to adding different concentrations of the NP treatment. The monolayer was then vertically scraped using a SPLScar commercial Scratcher-0.5 (SPL Lifesciences, Pocheon-si, Republic of Korea, Cat. No. 201924). After this procedure, three additional washes were performed with 1× PBS buffer (2 mL) to remove any detached cells due to scraping. Finally, the culture medium was added to each well. The cells were then incubated with CS NPs at a concentration of 100, 200, 300, or 400 μg/mL. Photographic monitoring was performed using optical microscopy at 4 h intervals until the wound was completely closed.

### 2.9. Inmunofluorescence

Cells (6 × 10^4^) were seeded on microscope slides, and the wound closure assay was carried out following the procedure described in Section 2.8. Subsequently, the cells were fixed with 4% paraformaldehyde at 37 °C for 30 min. The slides were washed three times with 1× PBS buffer to remove paraformaldehyde residues and stained with Phalloidin-FITC at a ratio of 1:250 at 37 °C for 1 h. Cell nuclei were then stained with Hoechst at a 1:1000 dilution at room temperature for 15 min. After this, coverslips were mounted over each slide with VECTASHIELD and visualized using confocal microscopy on a Carl Zeiss LSM900 (Carl Zeiss, Jena, Germany) microscope with a 10× objective. The obtained images were processed using ZEN 2.3 Pro Software.

## 3. Results and Discussion

### 3.1. CS NPs Formation and Structural Analysis

To form CS NPs, bulk chitosan was combined with sodium tripolyphosphate (TPP), which reacts directly with the functional groups in the chitosan molecule, such as amine and hydroxyl, forming inter- and intramolecular bonds with the phosphate groups of TPP (oxygen and phosphorus atoms) [20]. By varying the TPP concentration and stirring time, we obtained three average particle sizes: 15, 30, and 125 nm, as shown in Table 1. Figure 1a shows an AFM micrograph of nanoparticles with a particle size of <125 nm>. The scale bar indicates that almost all particles are less than 250 nanometers in diameter, with a maximum height of 18 nm. According to the three protocols described above (see Table 1), the nanoparticle (NP) size varied to include small NPs (~15 nm) that can cross the cell membrane and larger NPs (~125 nm) to ensure the presence of extracellular particles. Appendix A provide micrographs and a grain analysis of the corresponding protocols for <15 nm> and <30 nm>. Artifacts due to substrate relief were avoided by using an atomically flat substrate (silicon) with a nominal roughness of 5 Å or less. To validate the average size of the mentioned protocols as determined by AFM and TEM analysis, DLS characterization was performed, and the results are presented in Appendix A.

It is worth mentioning that the intention of TEM characterization was not to provide a realistic size distribution for the CS NPs, but to demonstrate the effect of our experimentally chosen values on the main variables of the well-known ionotropic gelation process. We realize that crystalline nanoparticles can be obtained, as we can observe from Figure 1b, where a well-defined elliptical nanoparticle (taking into account the AFM peak-to-peak information we can say about a nano disk) can be observed with a “major axis” beyond 10 nm; by doing a zoom of this nanoparticle, in Figure 1c, we can observe clearly defined crystalline planes; an electronic close-up was done for shown the measurement of the interplanar distance and 2.2 Angstroms was obtained, this is correlated with a diffraction angle of 44 degrees by employing Bragg relationship. By using the microscope soft Digital Micrograph (TM) 3.7.0, a Fourier transformation was done over Figure 1c, and as a result, at least two satellite points (red arrow marks) can be clearly distinguished in Figure 1d. Remember that the Fourier transformation will situate us in the reciprocal space. Thus, each point is a representation of a crystalline plane in the reciprocal space. Unfortunately, there is not enough information (powder diffraction files) on the crystalline planes of CS NP to indicate the planes shown. Crystalline properties have already been investigated by several other researchers but not always achieved on CS NPs. Those who succeed in it also use ionotropic gelation and a similar percentage of (TPP) to get their CS NPs [21]. In addition, there is at least one reported work that developed CS NPs produced by Penaeus semisulcatus, which showed HRTEM measurements with observable Moiré patterns (indicating crystalline planes), although the authors provided no further discussion of the figure [22].

### 3.2. Raman Spectroscopy

It is well established that Raman spectroscopy is a helpful tool for investigating the phase materials after nanoparticle synthesis. Then, we performed optical characterization using Raman spectroscopy to obtain the vibrational modes of the CS NPs. The Raman spectrum for CS NPs in Figure 2a exhibits several peaks at 479, 800, 914, and 1114 cm^−1^, which become more intense as the average nanoparticle size increases. The bulk chitosan exhibits only one intense peak at 1114 cm^−1^, which at least one reference establishes is related to the region of ν(C-O-C) vibrations [23]. Concerning the analysis of nanoparticle vibrational modes, it is traditionally expected that the spectra of nanoparticles follow those of bulk materials, as seen in published results elsewhere [24]. However, in our results, the only “normal mode” is the one corresponding to δ(COC) at 479 cm^−1^. Thus, the modes at 800 and 914 could be modes resulting from the nanoparticle formation, which means that nanoparticles have some different vibrational modes than “bulk” first because the translational symmetry of crystalline bulk materials is broken at grain boundaries, which results in the appearance of a surface and additional vibrational contributions from the interface [25]. The last sentence allows us to hypothesize that our Raman spectra of CS NPs differ from the “bulk” due to a certain degree of crystalline arrangement that can be observed at the high-resolution analysis of the TEM micrographs.

### 3.3. UV-VIS Spectroscopy

UV-VIS absorption experiments were performed to demonstrate the formation of CS NPs. Figure 2b shows the experimental absorbance of the three preparations, revealing two small broad shoulders centered at 258 and 299 nm. The last one of these “peaks” has already been reported and is likely associated with the formation of CS NPs [26]. For comparison of our results with the absorbance graph in published data, it is essential to realize that CS NPs obtained through biosynthesis processing may have a less dispersed size distribution than those obtained through chemical processing, such as ionotropic gelation. As a result of dispersion in NPs’ size, “convolved” peaks can be observed as “shoulders” on the absorbance spectra. Other broad peaks were observed at 744 and 973 nm and could be attributed to absorption caused by the partial aggregation of CS NPs, similarly to the light absorption behavior of metallic nanoparticles when they aggregate [27].

### 3.4. Cell Viability Assay

Although chitosan is widely reported as being biocompatible, the literature presents mainly studies with microfibers and objects that are significantly larger than nanoparticles. Thus, for future biomedical applications, it is important to evaluate cell viability when chitosan NPs are co-cultured with cells and NPs become almost as small as the distance between two lipid hemilayers of the membrane protein, and then small enough to pass through the cell membrane (in particular, those related to the basolateral membrane and cytoskeleton where adherent and tight junctions are placed). A quantitative analysis of cell viability in canine kidney (MDCK) epithelia was performed by comparing the control groups with those treated with CS NPs of three average diameters (15, 30, and 125 nm), and the results are shown in Figure 3. Concentrations of 100, 200, 300, and 400 µg/mL were used in this study, trying to emulate the high concentrations commonly found in in vivo experiments in the literature [9,28]. The 24 h graph shows that viability tends to monotonically decrease (for all CS NPs groups: 15, 30, and 125 nm) as the concentration increases. The most obvious example is observed in the 30 nm group, with a 33% decrease at 400 µg/mL.

Furthermore, we can observe a similar decrease in viability versus concentration in the 48 h graph for the 125 nm group. However, for the 30 nm group (48 h), viability at the highest concentration decreases by only 24%. Overall, it is important to note that CS NPs at the maximal concentration suggest the onset of a cytotoxic effect on cells. An ANOVA (*n* = 9) performed using GraphPad Prism 8.0.2 revealed that some of the differences in viability between the control and treated cells (for the different size distributions: 15, 30, and 125 nm) were statistically significant (*p* < 0.0001) at specific concentrations. The analysis was performed by grouping nanoparticle size and two different incubation times.

We can observe that more significant results are obtained for the 48-h incubation time. This could be expected considering that the endocytosis process could become the main mechanism for the internalization of the CS NPs in the cells and has been reported as time-dependent for epithelial cells [29]. It is important to mention that proteomic analysis of proteins associated with the endocytosis process was conducted.

### 3.5. Protein Expression of MDCK When Co-Cultured with CS NPs

Considering the viability test results, total protein extracts were analyzed using SDS-PAGE to determine whether there was a change in protein expression patterns due to the co-culture of CS NPs with MDCK cells. Thus, two concentrations (200 and 300 μg/mL) of the average 125 nm CS NPs were used for the assay, as shown in Figure 4. An evidently thicker band at approximately 66 kDa and two additional bands between 31 and 45 kDa were observed in the lanes with CS NP treatment conditions. This overexpression could primarily be due to the interaction of the CS NPs with the cells during the incubation period. Therefore, a quantitative proteomic analysis was necessary to distinguish which proteins were overexpressed due to the interaction with CS NPs during co-culturing (it is important to note that several proteins could have the same molecular weight).

### 3.6. Proteomic Analysis

A quantitative proteomic analysis was performed to compare the total proteins from MDCK cells cultured with and without CS NPs. After false discoveries were removed, proteins were listed in order from the highest to lowest amounts (femtomoles, fmol) in the samples treated with CS NPs. To perform a bioinformatic analysis, the overexpressed proteins were selected based on a cutoff of 10 times the femtomolar concentration of the control sample. The resulting proteins were processed using the Gene Ontology (GO) and String databases.

From the proteomic analysis, we focused on two main biological processes: endocytosis as an interaction pathway of CS NPs and cells and the cytoskeleton organization of cells as a reference to cell migration.

Regarding endocytosis, it is important to notice that due to their size, nanoparticles have not previously been tested (by Nature) against cellular molecular receptors. Therefore, cells may lack specific receptors for nanoparticles, and interactions are likely nonspecific. However, a commonly reported mechanism for cellular interaction with “foreign bodies” is phagocytosis, which is primarily carried out by professional phagocytes. Epithelial cells, such as MDCK, are not professional phagocytes; under normal conditions, they primarily perform endocytosis rather than phagocytosis. Among the various endocytic mechanisms, our proteomic data reveal that clathrin is overexpressed compared to control samples, suggesting that clathrin-mediated endocytosis (CME) may be the primary route for internalizing chitosan nanoparticles (CS NPs) in MDCK cells. However, further investigation by using Chlorpromazine (CPZ), which dissociates clathrin from the surface membrane, inhibiting endocytosis, will be required [30]. Additional proteins to clathrin, which are related to the endocytosis process, were found in our proteomics and are listed in Table 2. A comparison between the two different incubation times was shown. We found, for example, an under-regulation of caveolin, which is well known for its functions in caveolae, in apical endocytic recycling compartments on polarized epithelial cells [31].

Once internalized, CS NPs have the potential to interact with the entire host proteome. Importantly, to the authors’ knowledge, no studies have reported proteolytic activity on CS by the MDCK proteome.

Regarding cytoskeletal organization, the String analysis shows a network of interactions between proteins (Figure 5a and Figure 6a); we highlight proteins participating in cytoskeleton formation with red halos, and Table 3 was formed based on it. The overexpressed proteins were classified into biological processes (Figure 5b and Figure 6b), molecular components (Figure 5c and Figure 6c), and cellular functions (Figure 5d and Figure 6d). As mentioned above, the GO analysis showed that CS NPs favor the organization of the cytoskeleton.

Additionally, the proteomic data revealed that six proteins present in the experiments were absent from the control sample. By identifying each protein, we found a protein complex formed by ezrin (*EZR*), radixin (*RDX*), and moesin (*MSN*), which connects the actin cytoskeleton to the plasma membrane and regulates cell migration [32,33], cell motility, and T cell migration [34]. This result strongly suggests that chitosan contributes to the structuration of the cell cytoskeleton, even without the stimulus of wound production.

### 3.7. Wound Closure Assay

Following the interpretation of proteomics data, functional assays were performed to validate the GO analysis predictions, thereby confirming a direct association between cell migration and cytoskeletal organization. The induction of cell migration caused by CS NPs was verified using a wound closure assay. Preliminary experiments revealed that CS NPs accelerate cell displacement, and the <125 nm> nanoparticles had a better effect compared to those with average sizes of 15 and 30 nm; at Appendix A we can observe experimental results that proves our point for <15 nm>; thus, <125 nm> nanoparticles were employed for these assays. Figure 7a quantifies the width of each wound in µm (cell-free area) for the control and treatments with different concentrations of CS NPs at 0, 4, and 8 h, respectively. Notably, complete wound closure was observed before 8 h of treatment with NPs; for that reason, there is no bar at this bin. However, the same migration behavior was not observed for the control condition, where 45% of the empty cell area was still observed at 8 h. A two-way ANOVA was conducted to evaluate the effects of concentration and time on gradual wound closure. Using a significance level of 0.05, the analysis revealed that only the effect of time was statistically significant; detailed calculations are included in the Appendix A.

Figure 7b presents an immunofluorescence image of wound closure obtained using confocal microscopy, clearly showing the migration process in the presence of chitosan. For the demonstration, the highest concentrations of CS NPs were used (300 and 400 μg/mL). Appendix A show representative images of bright fields (Figure 7a) and the immunofluorescence (Figure 7b) of the wound closure assay. Furthermore, by analyzing the wound edge, we observed a morphological change in the cells. Actin protrusions, including filopodia and lamellipodia, are activated when the cell enters the migration phase (Figure 7c). These structures are essential for cell motility, and a list of one of the overexpressed proteins (Rho GTPases) has been reported as participating in filopodia formation as a consequence of stimulation [35]. In this work, the CS NPs cause the same response. Moreover, we observed the polymerization of actin fibers in the samples containing CS NPs. The migration results show that chitosan promotes a phenotypic change in MDCK cells due to protein enrichment, as demonstrated by the mass spectrometry results in Table 3, which lists proteins related to cytoskeletal formation. It is noticeable that most of the proteins are overexpressed; however, in the 48-h assay, their concentration remains below the detection limit of the technique.

Therefore, even though mass spectrometry is a reliable quantitative technique, it is traditional to observe a few of the overexpressed proteins using immunodetection assays to “prove” the biological function after any treatment. Moreover, further investigations will be needed to establish a molecular site at which CS NPs attach membrane proteins of the cell (according to our data, at least one membrane protein was found to be overexpressed, the NaK pump ATPase); future studies should perform molecular docking, and the results should be verified by the molecular genetics of the involved proteins.

An example of the previous statement for one of the proteins cited by our Table 3 can be found in the literature [36]. In this work, the authors showed that Annexin A (AnxA2), which is reported as an actin-binding protein, has an essential participation in the Intestinal Epithelial Cell Migration, stating that it “is up-regulated in migrating intestinal epithelial cells (IECs) and plays an important role in promoting wound closure”. However, they need to downregulate AnxA2 using two different shRNA sequences (shAnxA2) and a control cell line with a non-silencing shRNA target (shCtrl). In this work, we present the overexpression of Annexin 1 in Table 2. Additionally, Annexin 2 was also overexpressed, albeit at a low level (only 6 femtomols), due to the interaction of our epithelial model with CS NPs. Thus, we can hypothesize that our epithelial model is also capable of cell migration due to the coincidence in the regulation of Annexin; however, this conclusion lacks experimental proof, so the only valid conclusion at this moment is that further work is necessary for in vitro verification of the interactome shown in Figure 5a.

## 4. Conclusions

We successfully synthesized and characterized the physicochemical properties of CS NPs obtained through ionotropic gelation. Nanoparticles with three different reproducible average sizes exhibited no significant cytotoxicity when interacting with MDCK cells at concentrations below 400 μg/mL and for less than 24 h. Moreover, cell migration was observed during the closure of scratch assays when 125 nm CS NPs were added. Furthermore, migration is directly related to the overexpression of several proteins (like Anexin) involved in the actin cytoskeleton. Although the proteomics of MDCK have been identified, there is still a lack of explanation for the mechanism by which chitosan regulates all the found proteins.

## Figures and Tables

**Figure 1 molecules-30-03815-f001:**
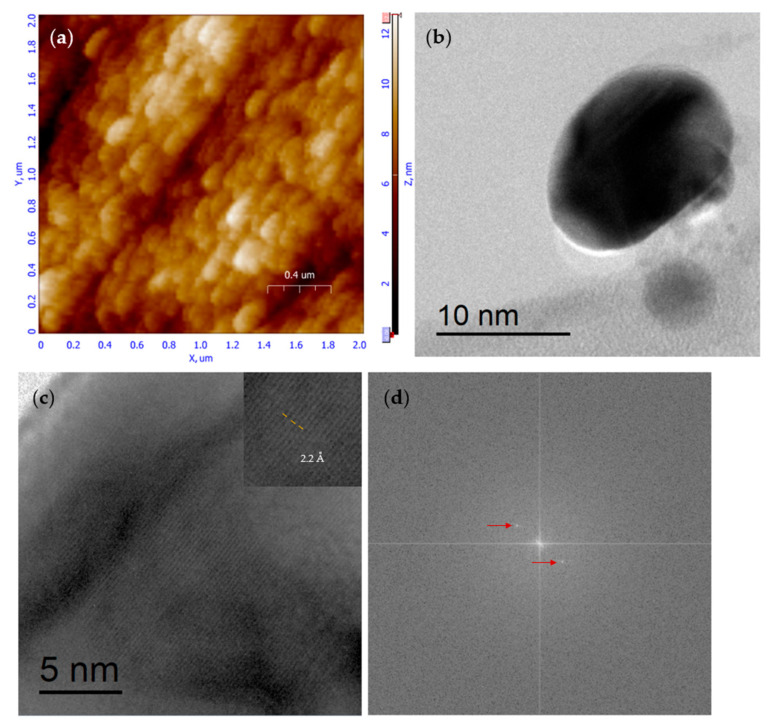
(**a**) AFM micrograph of 2 µM CS NPs with an average size of 125 nm after 30 min of stirring. (**b**) TEM image of CS NPs; (**c**) crystalline planes with an interplanar distance of 2.2 Å; and (**d**) Fourier transform of the micrograph shown in (**c**).

**Figure 2 molecules-30-03815-f002:**
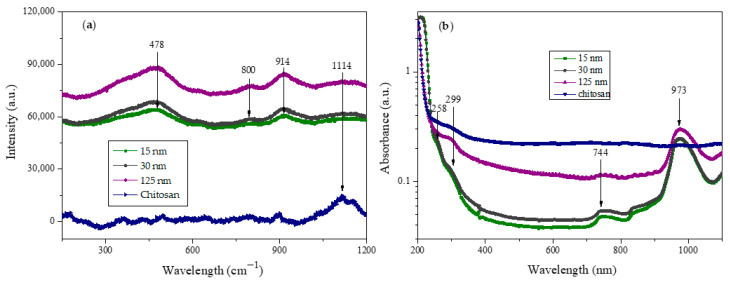
(**a**) Raman spectra obtained from raw, unprocessed chitosan and differently sized CS NPs. (**b**) UV-vis absorption spectra of different protocols of CS NPs.

**Figure 3 molecules-30-03815-f003:**
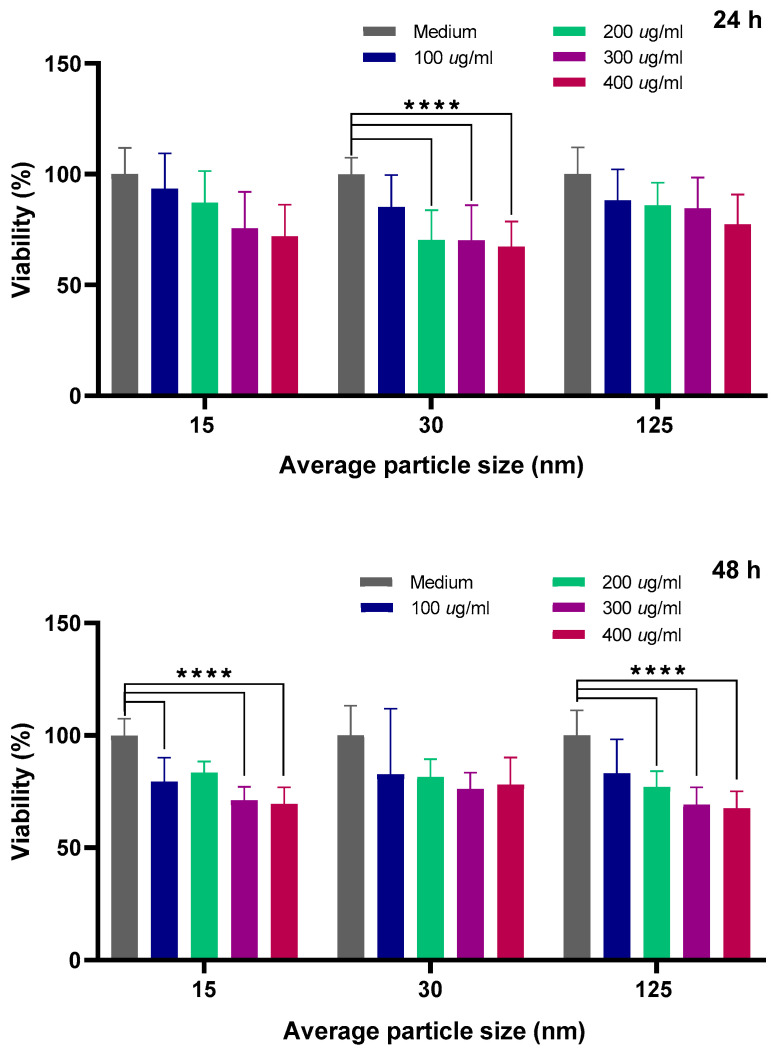
Percentage of cell viability of MDCK cells after treatments with CS NPs for 24 and 48 h at 37 °C. Results are shown as pooled data from three biological replicates of each nanoparticle size (15, 30, and 125 nm) and concentration with their respective statistical significance (****). Statistical analysis was performed with a significance level of 0.05.

**Figure 4 molecules-30-03815-f004:**
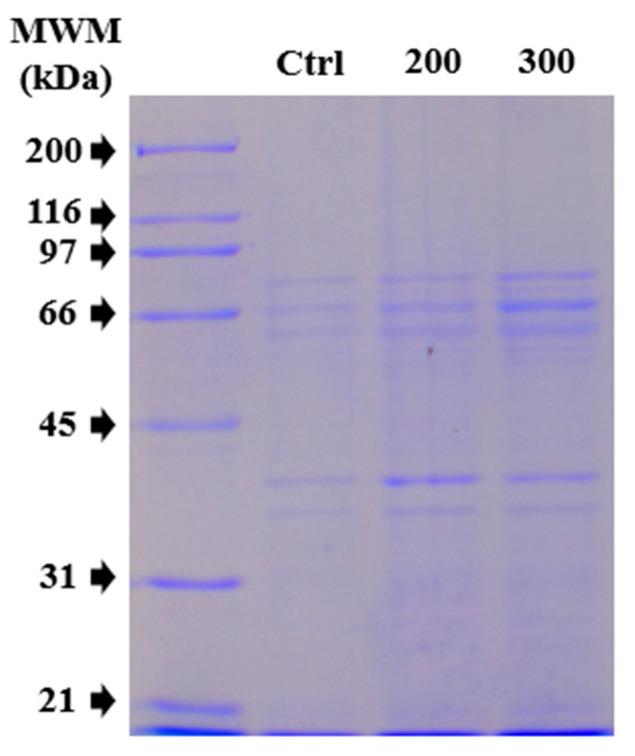
Coomassie blue staining of 10% SDS-PAGE of CTL and treatments with 125 nm CS NPs at 8 h.

**Figure 5 molecules-30-03815-f005:**
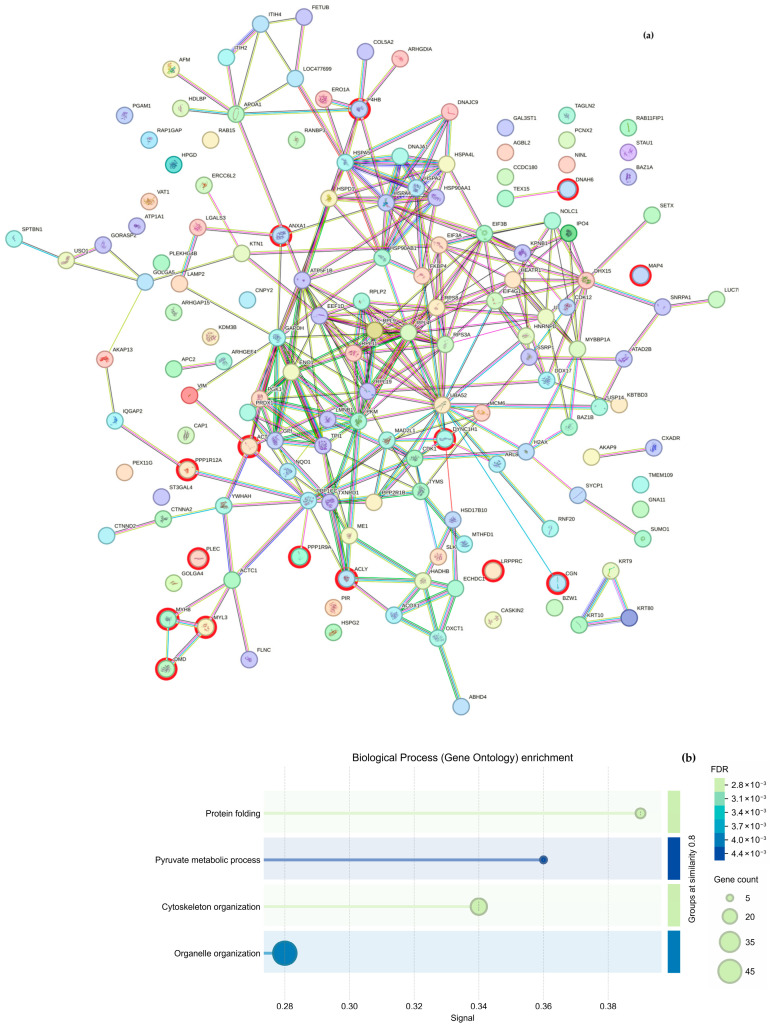
Proteomic from samples incubated with CS NPs for 8 h. (**a**) The interactome, obtained from STRING, highlights the interactions among proteins based on their 10×
overexpression. The results of the GO analysis of genes that participate in biological processes (**b**), molecular functions (**c**), and the cellular components (**d**) of the *Canis lupus familiaris*.

**Figure 6 molecules-30-03815-f006:**
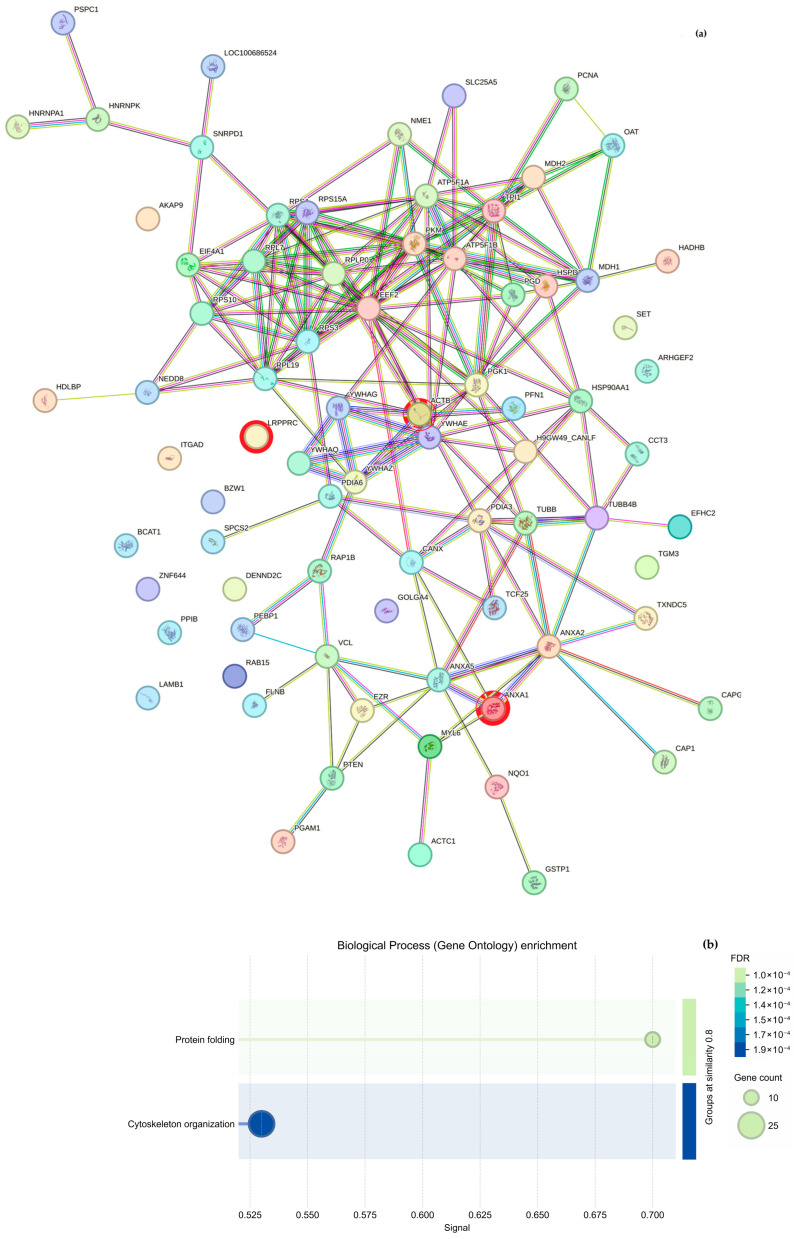
Proteomics from samples incubated with CS NPs for 48 h. (**a**) The interactome, obtained from STRING, highlights the interactions among proteins based on their 10×
overexpression. The results of the GO analysis of genes that participate in biological processes (**b**), molecular functions (**c**), and cellular components (**d**) of the *Canis lupus familiaris*.

**Figure 7 molecules-30-03815-f007:**
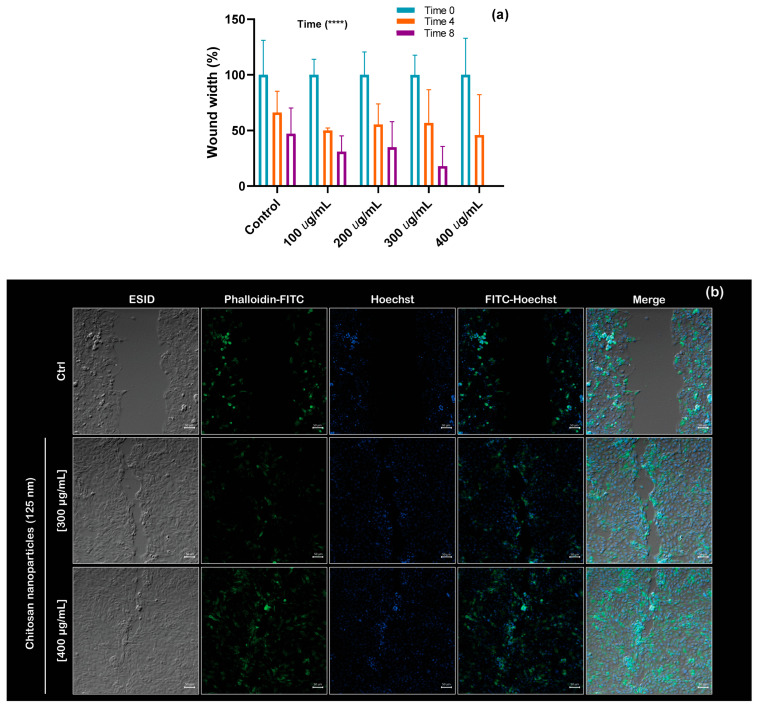
Cell migration induced by CS NPs. (**a**) Graph of wound closure kinetics at 0, 4, and 8 h, as well as the ANOVA analysis with a significance of *p* < 0.0001 (****). (**b**) Images obtained by confocal microscopy (10×) of the wound closure assay. (**c**) Morphological image (40×) of lamellipodia and filopodia, indicated by yellow arrows. For immunofluorescence, cells were stained with Phalloidin coupled with FITC to visualize the actin cytoskeleton and Hoechst to stain cell nuclei, with their respective merge images. Additionally, brightfield images (ESID) and coupling between ESID, cytoskeleton, and nucleus were also captured.

**Table 1 molecules-30-03815-t001:** Nanoparticle synthesis parameters.

Nanoparticle Size (nm)	Stirring Time	TPP Concentration (mg/mL)
<15 nm>	120 min	5 mg/mL
<30 nm>	120 min	0.8 mg/mL
<125 nm>	30 min	5 mg/mL

**Table 2 molecules-30-03815-t002:** Identified proteins related to endocytosis in MDCK cells.

Protein Name	Gene	Mass (kDa)	Control at 48 h (fmol)	CS NPs at 48 h (fmol)	Control at 8 h (fmol)	CS NPs at 8 h (fmol)
Annexin	* ANXA1 *	38	216	392	198	268
RAB15_member RAS oncogene family	* RAB15 *	48	141.6	318	109.7	188.7
Actin_cytoplasmic 1	* ACTB *	42	214.7	298	243	273
Adenylyl cyclase-associated protein	*CAP1*	51	21.8	40.8	29.2	46
Clatrhin heavy chain	*CLTCL1*	226	15	24.2	-	-
Heat shock protein HSP 90-beta	*HSP90AB1*	80	143.7	152.8	106	126
Rho GTPase activating protein 15	*ARHGAP15*	27	237.2	241	113.3	289
Heat shock-related 70 kDa protein 2	*HSPA2*	70	12.5	15.1	6.9	22.4
CXADR Ig-like cell adhesion molecule	*CXADR*	40	18.5	20.6	19	36.3
Lysosomal associated membrane protein 2	* LAMP2 *	45	-	-	39.6	145.9
Dynein cytoplasmic 1 heavy chain 1	* DYNC1H1 *	535	-	-	45.7	118.8
RAB11 family interacting protein 1	* RAB11FIP1 *	137	-	-	17.1	65
Pirin	* PIR *	32	-	-	39.7	85.2
Vimentin	* VIM *	53	-	-	62	80.8
Catenin alpha-2	* CTNNA2 *	101	-	-	12.3	29.2
Caveolin-1	* CAV1 *	20	188.7	65.4	53.6	47.8

**Table 3 molecules-30-03815-t003:** Identified cytoskeletal proteins in MDCK cells.

Protein Name	Gene	Mass (kDa)	Control at 8 h (fmol)	CS NPs at 8 h (fmol)	Control at 48 h (fmol)	CS NPs at 48 h (fmol)
Protein disulfide isomerase	*P4HB*	57	79.1	99.5		
Plectin	*PLEC*	539	10.8	18		
Leucine-rich pentatricopeptide repeat containing	*LRPPRC*	159	90.3	113.5	92	114.7
ATP citrate synthase	*ACLY*	118	47	73.1		
Dynein cytoplasmic 1 heavy chain 1	*DYNC1H1*	535	45.8	118.9		
Dynein axonemal heavy chain 6	*DNAH6*	480	59.5	139.2		
Actin_cytoplasmic 1	*ACTB*	42	250	273.1	214.8	298
Myosin-8	*MYH8*	223	96.3	181.7		
Dystrophin	*DMD*	427	2.8	25.3		
Annexin	*ANXA1*	38	198	268	216.2	392
Protein phosphatase 1 regulatory subunit	*PPP1R12A*	121	22.3	59		
Microtubule-associated protein	* MAP4 *	102	7.6	28.9		
Protein phosphatase 1 regulatory subunit 9A	* PPP1R9A *	151	100.5	321.8		
Myosin light chain 3	* MYL3 *	23	162.2	357.2		
Cingulin	* CGN *	136	42	82.3		

## Data Availability

The data are contained within the article and its Appendix A.

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
