# Peer review of "Proteomic Insights into the Interaction of Chitosan Nanoparticles with Canine MDCK Epithelial Cells"

_molecules, 2025, doi:10.3390/molecules30183815_

Round 1
Reviewer 1 Report (Previous Reviewer 2)
Comments and Suggestions for Authors
The manuscript under revision “On the overexpression of proteins from MDCK epithelial cultures under Chitosan nanoparticles treatments” presents well-supported data on the effect of chitosan nanoparticles on epithelial protein expression and migration. The revised manuscript shows substantial improvements in structure, clarity, data presentation, and analytical depth. The authors have responded effectively to reviewer concerns. I consider the manuscript is now suitable for publication after Minor Revisions, particularly in language polishing and clearer linking of overexpressed proteins with biological pathways.
Author Response
Dear reviewer, we appreciate your comments, which guide us to follow the highest quality in scientific research and its communication.

Reviewer 2 Report (Previous Reviewer 3)
Comments and Suggestions for Authors
The manuscript entitled “On the overexpression of proteins from MDCK epithelial cultures under Chitosan nanoparticles treatments” have prepared chitosan nanoparticles (CsNP) with different sizes using sodium tripolyphosphate and studied their interaction with Madin-Darby canine kidney (MDCK) Cells. The authors have reported that 3 average sizes of CsNPs were not toxic at the concentration tested. The authors have also studied the wound healing of CsNP by scratch assay as well as protein expression induced by presence of CsNPs in MDCK cells. CsNPs have been investigated in many research works which their cytotoxicity, wound healing and drug delivery have been studied. Besides, the authors mentioned that they have synthesized 3 CsNPs with different average sizes of 15, 50 and 150 nm. This study lacks providing enough evidence for the synthesis of CsNPs. The authors mentioned that they used 0.22μm filter for separating the particles smaller than 220 nm. The AFM images shown in the supplementary materials don’t show a clear image of the nanoparticles. TEM is provided only for one nanoparticle. To my opinion, the results do not conform with the past literature. Dynamic light scattering data are mandatory for particle size analysis; this is not reported in this manuscript. The grain analysis provided in the supplementary materials (S2) is wrong, the average size 0.540 μm is equal to 540 nm not 50 nm but the authors considered this as 50 nm. The statistical analysis of the data is not provided. In some figures, data are incomplete like fig. 6-a Time 8 is not presented for the sample 400 μg/ml. The manuscript is not well written. The latest research works on the synthesis of CSNP are not reviewed in introduction. In some places the text is not placed in the correct section, for example lines 362-365 belong to the experimental section. Another issue is that CSNPs are not used directly for wound healing due to lack of sufficient antibacterial activity over wide range of pathogens. The conclusions of this research do not contribute to the advancement of science in the field of wound healing using CsNPs. According to the above, I do not recommend this manuscript for publication in the Molecules journal.
Best Regards
Author Response
Dear reviewer, we appreciate your comments, which guide us to follow the highest quality in scientific research and its communication.

Reviewer 3 Report (New Reviewer)
Comments and Suggestions for Authors
- The Introduction part should be improved and updated with the recently published papers. There are numerous research and review articles dealing with similar or related topics. Furthermore, it is important to clearly define the aim of this research and emphasize what is new compared to previous studies.
- Line 92, Table 1……. It is more appropriate that explanations such as “nanoparticle size (nm), stirring time (min) and TPP concentration (mg/mL)” be given in the table itself instead of “A, B and C”, rather than in the table caption.
- Lines 96-98 …… In this sentence it is mentioned “behavior in metallic nanoparticles”. It remains unclear which metal nanoparticles are mentioned here, when there are no metal nanoparticles anywhere in this research.
- Line 108 ……. It is unclear how long samples were dried before AFM and Raman examination, 15 min or 1 hour?
- Lines 109-114 …….. Descriptions of UV-Vis and Raman characterization are mixed and confusing. To be clear, it would be better to describe one technique and then the other.
- Lines 195-197 ……. The sentence “Three average sizes of particles were obtained: 15, 50, and 125 nm as a result of varying the stirring time and TPP concentration from 120 to 30 min, respectively, as shown by Table 1” must be revised.
- Why was the text divided into parts 3.1, 3.2 and 3.3 when each part describes the same thing - CS NPs? It can be one section where CS NPs are described.
- Lines 205-206 ……. Figure 1a. presents AFM micrograph of 2 μM CS NPs with an average size of 125 nm obtained after 30 min stirring. It was stated that “The scale bar indicates that almost all particles are less than 250 micrometers in diameter, with a maximum height of 18 nm.” In the Supplementary, there is the grain size analysis of CS NPs which reveals that the average particle size was around 15, 50 and 125 nm. According to such a presentation of results, it is unclear what the dimensions of the CS NPs were. I think that the authors should pay extra attention so that such typos do not happen and call into doubt the reported results. From Figure 1a and the given scale-bar it is obvious that CS NPs are less than 250 NANOMETERS in diameter.
- The part that describes TEM analysis must be revised. It needs to be clarified whether the crystal planes are observed in only one direction (as stated in the text) or in two directions, as shown in the micrograph, because two arrows are represented. In addition, in my opinion, the TEM micrograph does not have the necessary quality and high enough resolution to unequivocally confirm the formation of single crystals.
- Line 225 ……. This is the first time that the authors mentioned “crosslinking agent”. What is the crosslinking agent in this research? Is it sodium tripolyphosphate (TPP)? Such data must be specified in the manuscript.
- Lines 227-229 …… The sentence “The bulk chitosan exhibits only one intense peak at 1114 cm-1 due to the loss of structural integrity when the size and dimensions of the chitosan are reduced from bulk to nanoparticles.” must be revised. There is literature data on what vibrations correspond to the band at around 1114 in the Raman spectrum of chitosan. In addition, the overall Raman analysis must be more clearly presented.
- Lines 235-237 ……. Ref. 17 does not confirm the claims stated (in Ref. 17 there is no shoulder in the spectrum, but a clearly visible peak), while Ref. 18 described completely different research (it deals with the gold nanoparticles, not CS NPs). Moreover, Ref. 22 is not mentioned anywhere in the manuscript.
- In the part of the manuscript that refers to the results and discussion, there are only a few references and a lot of assertions and conclusions. It is necessary to corroborate or connect the conclusions based on the obtained results with already published research and to add appropriate references to the manuscript.
Author Response
Dear reviewer, we appreciate your comments, which guide us to follow the highest quality in scientific research and its communication.

Reviewer 4 Report (New Reviewer)
Comments and Suggestions for Authors
In the manuscript entitled “The overexpression of proteins from MDCK epithelial cultures under Chitosan nanoparticle treatments” the authors reported a proteomic analysis of chitosan nanoparticles (NPs)-treated MDCK cells. The protocol used to synthesize the NPs is standard, and the methods used for physicochemical characterization were in line with the literature. Despite the proteomic investigation of chitosan NP effects on MDCK not having been reported yet, the significance of the results is not completely convincing. There are several aspects that must be considered by the authors before the manuscript is suitable for publication.
- I suggest changing the title. I report a putative title “Proteomic insights into the interaction of Chitosan nanoparticles with canine MDCK epithelial cells”
- The authors must revise the introduction in line with the reported results. The investigation of cell migration with the scratch assay is not sufficient to determine the wound healing ability of the investigated NPs. The focus of the manuscript should be directed to the detection of the molecular mechanism modulated by chitosan NPs. In this respect, the first paragraph is not necessary; in contrast, the authors should better describe the literature on chitosan NPs and how they can be used in different biological applications.
- Line 303-304, please introduce a literature reference in support of the sentence
- Regarding the cell viability results, the authors should clarify the type of ANOVA analysis that was performed, and if all the reported lines in Figure 3 refer to the **** significance mark. Please also introduce the explanation for **** in the figure legend. Furthermore, the authors should discuss in the results section how they can explain that the cell viability was significantly decreased after 24h treatment with 30 nm NPs (with a significance of ****) and no significant difference was observed after 48h. How is it possible? The authors should consider improving the number of experiments to obtain more consistent results.
- The authors should better define why they decided to use a 125 nm NP size for further experiments instead of 15 nm, considering that they presented the same trends in the cell viability assay.
- Regarding the scratch assay, the authors must perform an appropriate statistical analysis to determine the significance of the results (considering the presence of two variables, time and concentrations)
- Line 309: should be 30 nm, not 50 nm
Author Response
Reviewer,
In the manuscript entitled “The overexpression of proteins from MDCK epithelial cultures under Chitosan nanoparticle treatments” the authors reported a proteomic analysis of chitosan nanoparticles (NPs)-treated MDCK cells. The protocol used to synthesize the NPs is standard, and the methods used for physicochemical characterization were in line with the literature. Despite the proteomic investigation of chitosan NP effects on MDCK not having been reported yet, the significance of the results is not completely convincing. There are several aspects that must be considered by the authors before the manuscript is suitable for publication.
- I suggest changing the title. I report a putative title “Proteomic insights into the interaction of Chitosan nanoparticles with canine MDCK epithelial cells”
- Dear Reviewer, thanks for the suggestion. It has been done.
- The authors must revise the introduction in line with the reported results. The investigation of cell migration with the scratch assay is not sufficient to determine the wound healing ability of the investigated NPs.
- Dear Reviewer. We understood the point and attempted to rewrite the introduction section to emphasize that the primary objective is to comprehend the molecular interactions (revealed by proteomic assays) between CS and epithelial cells, rather than the wound healing properties commonly reported for animal models.
- The focus of the manuscript should be directed to the detection of the molecular mechanism modulated by chitosan NPs.
Dear Reviewer, thank you for your thoughtful suggestions. Indeed, this question is part of our long-term methodological study, which will require multiple molecular biology experiments to investigate the majority of the overexpressed and underexpressed proteins identified in our proteomics analysis. At a preliminary level, however, we observed the overexpression of Rab15 at the 8-hour mass spectrum. This protein belongs to the Rab GTPase family (detected in our raw data as Ras oncogene family, member 15).
It is well established that proteins such as coat proteins, adaptors, retrieval proteins, and scission proteins regulate endocytosis and intracellular trafficking, with Rab GTPases playing a central role in these processes. Therefore, the overexpression of Rab15 may represent a potential indicator of endocytic activity.
To validate this hypothesis, a knock-out analysis would be required. However, such an investigation lies beyond the scope of the present work.
Moreover, some other proteins appear at our proteomic data, that weren´t considered due to its “low” overexpression (below 1 femtomole): SCR kinase signaling inhibitor 1 (SRCIN1), which is part of the Src family kinases, which are involved in regulating various aspects of endocytosis, and even a recent article mention it as participating on Macropinocytosis at the Apical Surface of Polarized MDCK [2].
For this reason, the text was modified consequently, and an additional table with potential proteins participating in endocytosis was included.
[1] https://doi.org/10.1016/S0896-6273(00)80664-9
[2] https://doi.org/10.1111/j.1600-0854.2006.00412.x
In this respect, the first paragraph is not necessary; in contrast, the authors should better describe the literature on chitosan NPs and how they can be used in different biological applications.
Dear reviewer, thank you for your kind suggestion. The introduction has been modified according to your suggestion. However, the old reference to the pioneering work on wound healing was maintained, as we believe it is necessary to emphasize the importance of further studies on epithelia other than skin.
- Line 303-304, please introduce a literature reference in support of the sentence.
Thanks for the recommendation. The literature was reviewed, and a relevant reference was included.
- Regarding the cell viability results, the authors should clarify the type of ANOVA analysis that was performed, and if all the reported lines in Figure 3 refer to the **** significance mark. Please also introduce the explanation for **** in the figure legend.
Thanks for the observation; the changes were made to the text and highlighted with a yellow marker.
- Furthermore, the authors should discuss in the results section how they can explain that the cell viability was significantly decreased after 24h treatment with 30 nm NPs (with a significance of ****) and no significant difference was observed after 48h. How is it possible?
Thanks for your observation. At 48 hours and 100 μg/ml, the data had a high standard deviation, which explains why the result was not statistically significant. Moreover, we attempted to examine the literature looking for a time-dependent reference for the internalization of CS NPs. The reference was included in the draft. We also include the proteomic analysis for 48 hours to emphasize that at least time is a variable that influences cell metabolism.
- The authors should consider improving the number of experiments to obtain more consistent results.
Thank you for your recommendation. We develop the experiment within the standard criterion of 3 biological replicas and each with 3 technical replicas. We consider that this is standard number of experiments for readers interested on perform a comparison with previous and future experiments reported in the literature. However, we can carry out this work if it is mandatory according to your requirements.
The authors should better define why they decided to use a 125 nm NP size for further experiments instead of 15 nm, considering that they presented the same trends in the cell viability assay.
Thanks for the recommendation. We observed from the wound closure assays that 15 nm CS NPs were less effective than 125 nm CS NPs.
We added as supplementary material micrographs to prove the point.
- Regarding the scratch assay, the authors must perform an appropriate statistical analysis to determine the significance of the results (considering the presence of two variables, time and concentrations).
- Thank you for your kind recommendation. A two-way ANOVA was performed; the main text has been updated accordingly, and detailed calculations have been added to the supplementary material.
- Line 309: should be 30 nm, not 50 nm
Thanks for the observation. The data was fixed.
Round 2
Reviewer 2 Report (Previous Reviewer 3)
Comments and Suggestions for Authors
to my opinion, this study doesn't provide enough evidence on the formation of chitosan nanoparticles with the mentioned size. I still do not recommend publication of this manuscript.
Author Response
We appreciate the comments that encouraged us to improve our work. At the time of the review, we had not yet performed DLS analysis. We later carried it out in order to provide scientific evidence to support local characterization techniques such as AFM and TEM, which are capable of analyzing only nanovolumes. We regret that our work did not meet your expectations.

Reviewer 3 Report (New Reviewer)
Comments and Suggestions for Authors
- The aim of the research and the novelties are still not clearly presented.
- Line 102: Although it was already mentioned in the first review, it was still not mentioned in the text that the TPP is used as a crosslinking agent.
- Lines 107-108: I would ask the authors to explain how filtering through the membrane sterilizes the samples. By filtering samples through a membrane, they are purified, while sterilization is something completely different.
- Lines 112 and 127: In the text, the authors mention the determination of the band gap for CS NPs. However, the band gap was not determined anywhere in the work.
- Line 252: It was stated “the average grain size increases”, but is more appropriate to say “the average nanoparticle size increases”.
- Lines 254-255: The sentence “Concerning nanoparticle vibrational modes analysis.” is unclear or is it incomplete.
- Line 287: It should state "absorption spectra" instead of "Absorbance spectrum".
- Line 302: It says "is observed in the 50 nm group", but the nanoparticles are not 50 nm, they are 30 nm.
Author Response
To referee three.
Dear reviewer, thanks for your valuable comments that guide us to improve the manuscript
- The aim of the research and the novelties are still not clearly presented.
We appreciated your kind observation to improve our work; thus, abstract was modified to include the aim of the research.
The objective of our study is to determine how is that CS NPs participate in the regulation of different proteins of the MDCK epithelial model.
In addition, the introduction was added with a final sentence (line 86):
Moreover, the proteomics developed in this work could be useful in determining which proteins could serve as a starting point for studying the signaling pathways involved in the capture and internalization of chitosan.
- Line 102: Although it was already mentioned in the first review, it was still not mentioned in the text that the TPP is used as a crosslinking agent. Thanks for the observation, it was included at line 89.
- Lines 107-108: I would ask the authors to explain how filtering through the membrane sterilizes the samples. By filtering samples through a membrane, they are purified, while sterilization is something completely different. Thanks, to clarity, the word was changed and the text was modified.
- Lines 112 and 127: In the text, the authors mention the determination of the band gap for CS NPs. However, the band gap was not determined anywhere in the work. Thanks for the observation, the word was exchanged by absorption, because only semiconductors have formally a band gap.
Line 252: It was stated “the average grain size increases”, but is more appropriate to say “the average nanoparticle size increases”. Thanks, it was done.
- Lines 254-255: The sentence “Concerning nanoparticle vibrational modes analysis.” is unclear or is it incomplete.
Thanks for the observation, the sentence was modified thus a completed idea was expressed.
- Line 287: It should state "absorption spectra" instead of "Absorbance spectrum". Thanks, it was modified in the text.
- Line 302: It says "is observed in the 50 nm group", but the nanoparticles are not 50 nm, they are 30 nm. We apologize for this mistake; it was modified in the text.

Reviewer 4 Report (New Reviewer)
Comments and Suggestions for Authors
The authors have implemented some of the requested revisions by modifying the text, while in other cases they have provided justification for not being able to comply. For these reasons, the manuscript can be considered acceptable for publication
This manuscript is a resubmission of an earlier submission. The following is a list of the peer review reports and author responses from that submission.
Round 1
Reviewer 1 Report
Comments and Suggestions for Authors The paper investigates the effects of chitosan nanoparticles (CsNPs) of different sizes on the migration and wound healing of MDCK cells, and the topic is of significant application value. The experimental design is systematic, combining physicochemical characterization, cytotoxicity, migration experiments, and proteomics analysis, with data supporting the main conclusions. However, the authors need to address the following issues to further enhance the quality of the paper:- The abstract mentions that “CsNPs have a crystalline structure with a diameter of less than 15 nm,” while the main text describes the synthesis of particles with sizes of 15, 50, and 125 nm. Is there a contradiction in the description? This needs to be clarified.
- The specific parameters for the synthesis of nanoparticles (such as temperature, stirring time, and the ratio of TPP to chitosan) should be provided, but these key parameters are not shown in the main text or examples.
- In the cell viability experiment, all sizes of CsNPs showed no toxicity at a concentration of 400 μg/mL. However, Figure 2 shows a significant decrease in survival rate (p<0.05) for the 125 nm group at 48 hours. Should this contradiction be discussed?
- The SDS-PAGE shows overexpression of a 57 kDa protein, but its direct association with migration has not been verified through Western blot or functional experiments.
- In Figure 3a, the “Control” group is not labeled in the legend, and some images (such as the immunofluorescence images) have insufficient resolution. These need to be optimized.
- The discussion on the mechanism by which 125 nm CsNPs promote migration (such as cytoskeletal reorganization and signaling pathways) is insufficient. It is recommended to conduct a more in-depth analysis in combination with the results of proteomics.
- Some references are incomplete. For example, Reference 15 is only labeled as “NCBI Gene Database,” and the full entry needs to be supplemented.
- There are grammatical errors in the main text (such as “femtomole concentration below our cutting criteria”) and inconsistent terminology (e.g., “fibopodia” should be “filopodia”). It is recommended to polish the entire text.
It is recommended to polish the entire manuscript.
Reviewer 2 Report
Comments and Suggestions for Authors
In this paper, "The study of the interaction of Chitosan NPs with epithelial cultures: a simplified scheme of wound healing at a cellular level", the authors present a new model to study the biocompatibility and action of CsNPs with different diameters using Madin-Darby canine kidney (MDCK) cells.
Although the study was carried out rigorously, there are many studies in the bibliography that abord the biocompatibility and wound healing mechanism of action. The paper fails to mention the importance of this study in relation to the existing data and finally does not reach the objective respect to give evidence of interaction between CsNPs and cells.
I considered that this manuscript was not suitable for publication in Molecules due to its lack of originality and mechanistic data.
Major Comments
- 2.5 Cell viability assay - Although the error bars and statistical analysis suggest that all results should be similar due to the high error bars in the figure, no significant differences are apparent between CsNPs of different sizes. Also, only concentrations below 400ug/mL were examined.
- The use of MDCK cells is valid for studying epithelial migration, but validation in a more physiologically relevant model such as human keratinocytes or fibroblasts would increase the translational impact of the findings.
- 2.6. Wound closure assay - in vitro wound healing assays have been performed up to 8 h. The data for this time at a concentration of 400 ug/mL are missing, the area is not zero due to the uncovered areas observed on the micrographs.
- Page 6. The authors claim that "Analysis of these results suggests that the increase in some proteins may be directly related to the biological processes of migration and wound healing; however, this suggestion requires further confirmation". However, in the title and abstract they propose to study the interaction of chitosan NPs with epithelial cultures. The lack of evidence and further studies left the manuscript without valuable information that could be included here. Furthermore, considering the high number of papers related to the use of CsNPs in wound healing.
- While the proteomic data indicate an upregulation of proteins related to cell migration and wound healing, the discussion does not sufficiently explore the molecular pathways that may be responsible for these effects. Inclusion of references to relevant signaling pathways (e.g. integrins, cadherins, MAPK) would improve the mechanistic understanding.
Comments on the Quality of English LanguageNo comments
Reviewer 3 Report
Comments and Suggestions for Authors
according to the attachment
